# Drought and Elevated CO₂ Impacts Photosynthesis and Biochemicals of Basil (*Ocimum basilicum* L.)

**T. Casey Barickman** [1,*] **, Bikash Adhikari** [1] **, Akanksha Sehgal** [2] **, C. Hunt Walne** [2] **, K. Raja Reddy** [2] **and Wei Gao** [3]

1 North Mississippi Research and Extension Center, Mississippi State University, Verona, MS 38879, USA; ba917@msstate.edu

2 Department of Plant and Soil Sciences, Mississippi State University, Mississippi State, MS 39762, USA; as5002@msstate.edu (A.S.); chw148@msstate.edu (C.H.W.); krreddy@pss.msstate.edu (K.R.R.)

3 USDA UVB Monitoring and Research Program, Natural Resource Ecology Laboratory, Department of Ecosystem Science and Sustainability, Colorado State University, Fort Collins, CO 80523, USA; wei.gao@colostate.edu

* Correspondence: t.c.barickman@msstate.edu; Tel.: +1-(662)-566-2201

**Abstract:** Drought-induced reduction in crop growth and productivity can be compensated by increasing atmospheric carbon dioxide ($CO_2$), a significant contributor to climate change. Drought stress (DS) affects crops worldwide due to dwindling water resources and irregular rainfall patterns. The experiment was set up under a randomized complete block design within a three-by-two factorial arrangement. Six SPAR chambers represent three blocks (10 replications each), where each chamber has 30 pots in three rows. Each chamber was maintained with 30/22 (day/night) °C temperature, with either ambient ($aCO_2$; 420 ppm) or elevated $CO_2$ ($eCO_2$; 720 ppm) concentrations. This experiment was designed to address the impact of DS on the physiological and biochemical attributes and study how the $eCO_2$ helps alleviate the adversity of DS in basil. The study demonstrated that DS + $eCO_2$ application highly accelerated the decrease in all forms of carotene and xanthophylls. $eCO_2$ positively influenced and increased anthocyanin (Antho) and chlorophyll (LChl). $eCO_2$ supplementation increased LChl content in basil under DS. Furthermore, DS significantly impeded the photosynthetic system in plants by decreasing $CO_2$ availability and causing stomatal closure. Although $eCO_2$ did not increase net photosynthesis (Pn) activity, it decreased stomatal conductance (gs) and leaf transpiration rate (E) under DS, showing that $eCO_2$ can improve plant water use efficiency by lowering E and gs. Peroxidase and ascorbate activity were higher due to the $eCO_2$ supply to acclimate the basil under the DS condition. This study suggests that the combination of $eCO_2$ during DS positively impacts basil's photosynthetic parameters and biochemical traits than $aCO_2$.

**Keywords:** chlorophyll; metabolites; carotenoids; antioxidants; phenotype

## 1. Introduction

Over the last few decades, the influence of global climate change on agricultural productivity has emerged as a critical research issue [1]. With continued projected modifications in climate and the need to ensure future food supply, understanding the effects of climate change on the productivity of major agricultural crop species has become critical [1,2]. Water resources are responsible for 80–95% of the fresh biomass of nonwoody plants and have a fundamental role in plant growth, development, and metabolism [3–5]. However, water resources are the most affected due to the rising linear temperature trend of 0.74 °C (1906–2005) over 100 years [5]. Plants are often subjected to numerous environmental stresses in natural and agricultural settings [6]. Since drying up water resources and erratic rainfall patterns are evident globally, this situation leads to drought stress (DS) to the crops [7]. Moisture storing capacity of the soil, rainfall distribution, and natural disasters are other factors that cause DS, and these factors also make the severity of DS

unpredictable [8]. DS is multifaceted stress that affects plants' physiological, morphological, biochemical, and molecular properties [9,10]. Although plants have evolved specialized acclimation mechanisms to respond to short and long-term DS to some extent [11], it is undeniable that DS affects plants in several ways. For example, a study in maize by Earl and Davis [12] stated that DS affected crop productivity in three different ways: (1) lowering the absorption of photosynthetically active radiation by crop canopy; (2) reducing the radiation use efficiency, and (3) limiting the harvest index.

On the other hand, DS increased the formation of reactive oxygen species (ROS), which includes superoxide and hydrogen peroxide ($H_2O_2$) [11,13,14]. These ROS are combatted by the plant antioxidant system (ascorbate, glutathione, and superoxide dismutase (SOD)) [13,14]. These antioxidants scavenged ROS in the following steps: (I) SOD converts $O_2^-$ into $H_2O_2$, and (II) ascorbate and glutathione remove $H_2O_2$ from the plant system [15].

On the other hand, rising atmospheric carbon dioxide ($CO_2$) may change the precipitation pattern in the long run, which is crucial for climate change. The change in rainfall pattern can directly impact soil temperature level and soil moisture content, significantly decreasing crop yield in the next 50 years [16]. Furthermore, the expected rise in drought conditions resulting from increased $CO_2$ and temperature in the atmosphere will also affect crop growth and production of basil (*Ocimum basilicum,* L.) [17]. Thus, it is crucial to evaluate essential climate drivers, the drought conditions, and the role of eCO$_2$ in basil growth and development to identify successful drought mitigation and adapt the crop to that climate.

Basil is the most widespread warm-season aromatic and medicinal herb, and it belongs to the subfamily Nepetoidae under the *Lamiaceae* family [18,19]. It is used as an ingredient for commercial fragrances and improves the shelf life of food products [20,21]. Since ancient times, it has been used for medicinal purposes, including treating headaches, lowering cholesterol, sugar, blood pressure, and kidney failure [22,23]. Basil contains several essential oils (1.5%), phenolic compounds, flavonoids, glycosides, and organic acids [24–27]. Basil thrives well under an optimal temperature range of 25–30 °C [28,29]. However, for commercial basil production to be efficient, supplemental water is needed. Plants react to DS via a series of physiological and biochemical responses [30,31]. Under DS, leaf water potential diminishes due to a higher transpiration rate than its absorption rate. These changes can result in the closure of the stomatal opening and a decrease in cell enlargement and growth [32].

Furthermore, DS inhibits cell elongation by reducing the turgor pressure and impairs cell division by reducing metabolism [33]. The photosynthetic rate of a crop is affected by DS as it causes a decrease in gas exchange activity and carbon assimilation [17,34,35]. The respiration and ion uptake are also reduced under DS, resulting in changes in the metabolic process and crop growth patterns resulting in crop failure [36]. For example, the reduction in photosynthetic rate, transpiration, and water use efficiency due to DS is also reported in several crops like basil [17], cowpea [31], maize [37], and mungbean genotype [38].

Furthermore, DS can also escalate reactive oxygen species responsible for oxidative stress that cause lipid peroxidation and alteration of both chlorophylls a (Chla) and b (Chlb) [9,39]. Chla and Chlb decreased in basil leaf tissue when basil plants were under DS and high-temperature stress for 14 days [40,41]. Similarly, superoxide dismutase (SOD), ascorbate, peroxidase, and antioxidant glutathione control the oxidative damage caused by reactive oxygen species ($H_2O_2$ anions) [13,42,43]. Previous research from Heidari and Golpayegani [42] demonstrated that hydrogen peroxide and superoxide anions increased in basil plants when subjected to DS.

According to a previous study, basil production increased by up to 80% in response to $CO_2$ levels rising from 360 to 620 ppm [44]. Studies have shown that eCO$_2$ enhanced the photosynthetic process and enriched the metabolites and the antioxidant activity in basil, parsley, and peppermint [44,45]. Al Jaouni et al. [44] also reported that eCO$_2$ (620 ppm) improved photosynthetic products and biomass accumulation by 40%. eCO$_2$ also promotes the proliferation of phenolics, flavonoids, glutathione, and several other antioxidants

to help combat damages from ROS during DS [44]. These dietary, physiological, and biochemical advantages of $eCO_2$ in basil may benefit human health concerns. Furthermore, screening stomatal conductance, photosynthetic rate, and water use efficiency under DS might help identify resistant genotypes. Thus, the current study's primary purpose is to understand the effect of DS coupled with $eCO_2$ on several physiological parameters, photosynthetic rates, carotenoids, chlorophylls, and several antioxidant concentrations in basil.

## 2. Results and Discussion

### 2.1. Physiological and Gas Exchange Measurements

Drought is one of the significant factors for damaging the photosynthetic pigments and thylakoid membranes [46]. DS inhibits the photosynthetic apparatus in plants by declining $CO_2$ availability and stomatal closure [35,47]. Several basil compounds have health benefits to humans, including chlorophylls, anthocyanins, flavonoids, and phenolics [48]. Leaf chlorophyll (LChl) content, epidermal flavonoids (flav), epidermal anthocyanin (antho), and nitrogen balance index (NBI) were measured as shown in Table 1. Antho and flav compounds together are responsible for antioxidant activity in the plants [49]. Flav are ubiquitous secondary metabolites in plants, which help protect the plant from abiotic and biotic stresses, while antho reduces the damage caused by free radical activity. [50]. Both antho and flav are reported to increase in different crops when subjected to DS + $eCO_2$ conditions [51–53]. However, flav content in the present findings contradicts many studies as flavonoid level was indifferent to the control under DS + $eCO_2$ condition [46,54,55]. Antho, on the other hand, decreased under DS + $aCO_2$ but increased under DS + $eCO_2$ as reported earlier [51,56]. Furthermore, NBI increased under DS + $aCO_2$ by 26.2% compared to control. NBI is the ratio of chlorophyll and epidermal flavanol [57]. DS decreases the nitrogen isotope composition and increases the accumulation of antho coupled with a transient decrease in LChl and NBI [54]. In the present study, LChl increased by 20% and 16% under DS + $aCO_2$ and DS + $eCO_2,$ respectively, compared to control (Table 1). It is determined that $eCO_2$ positively impacts and increases the NBI and LChl by alleviating the adverse effect of DS.

**Table 1.** The mean of chlorophyll (Chl), flavonoids (Fla), anthocyanin (Antho), and nitrogen balance index (NBI) of basil plants grown without drought stress (control) and with drought stress at two levels of $CO_2$ (420 and 720 ppm) after 17 days of treatment.

| Treatment | Chlorophyll [1] | Flavonoids | Anthocyanin | NBI |
|---|---|---|---|---|
| | [µg/mL] | [mg/g DM] | [mg/g DM] | |
| | | 420 ppm | | |
| Control | 21.468 bc | 0.685 ab | 0.114 b | 32.415 b |
| Drought | 25.744 a | 0.645 b | 0.102 c | 40.890 a |
| | | 720 ppm | | |
| Control | 18.978 c | 0.704 ab | 0.113 bc | 28.062 c |
| Drought | 22.027 b | 0.739 a | 0.127 a | 30.391 bc |
| Treatment [2] | *** | ns | *** | *** |
| $CO_2$ | ** | * | ** | *** |
| Trt*$CO_2$ | ns | ns | ns | * |

[1] Mean separation within the column by Duncan's multiple range test; ns, *, **, *** indicate non-significant or significant at $p \leq 0.05$, 0.01, 0.001, respectively; values followed by the same letter are not significantly different. Data are presented as means $\pm$ SE ($n = 10$). [2] SE: standard error of the mean, Chl = 0.900; Fla = 0.03; Antho = 0.004; NBI = 1.600.

The overall photosynthetic process is affected by several stress-induced stomatal limitations or metabolic impairment [55]. Under DS, stomatal conductance (gs) tends to be reduced temporarily, which affects the leaf transpiration rate (E) and intercellular $CO_2$ (Ci) assimilation [58,59]. In this study, no interaction effect ($p > 0.05$) was observed in net photosynthesis (Pn), gs, electron transport rate (ETR), and leaf temperature (Tleaf)

(Table 2). However, the treatment effect was observed in gs, where gs reduced significantly under DS in both $CO_2$ levels compared to control. Similarly, Ci lowered under DS + $eCO_2$, and DS + $aCO_2$ application decreased by 32.8% and 45.1%, respectively, compared to control. This decrement in Ci is due to reduced gs to prevent leaf water loss (wilting), as Saibo et al. [55] reported. Additionally, a treatment effect ($p < 0.001$) on Pn, was observed where there was a significant reduction of Pn under DS + $eCO_2$ (Table 2). A previous study reported that in the presence of $eCO_2$, Pn could increase along with the more activity of the rubisco enzyme and reduced photorespiration [60,61]. Therefore, in this study, $eCO_2$ cannot alleviate the negative effect of DS through Pn's increment. This decrement in Pn under DS is reported due to a reduction in gs by Saibo et al. [55]. Although $eCO_2$ could not increase Pn activity, the gs and E decreased under DS + $eCO_2$ compared to control (Table 2). This result is supported by several reports where $eCO_2$ application increases plants' water use efficiency by reducing the E and gs [62–64]. The intercellular/ambient $CO_2$ (Ci/Ca) ratio decreased significantly by 33% ($p < 0.001$) and 45% ($p < 0.001$) under DS followed by $aCO_2$ and $eCO_2$ application respectively. The reduction in Ci/Ca under DS suggests that the reduction in Pn could also be due to decreased Ci/Ca. This result was further supported by a report by Rajasekaran and Blake [65]. Even though $eCO_2$ cannot increase Pn in this study, $eCO_2$ application fulfills the gap created due to reduced Ci due to DS and decreased E in leaves to maintain water loss stated by Acock [66].

**Table 2.** The mean of net photosynthesis (Pn), stomatal conductance to water vapor (gs), intercellular $CO_2$ concentration (Ci), electron transport rate (ETR), leaf transpiration rate (E), leaf temperature (Tleaf), and intercellular/ambient $CO_2$ ratio (CiCa) of basil plants grown under without drought stress (control) and with drought stress at two levels of $CO_2$ (420 and 720 ppm) after 17 days of treatment.

| Treatment | Pn | gs | Ci | ETR | E | Tleaf | CiCa [1] |
|---|---|---|---|---|---|---|---|
| | | | 420 ppm | | | | |
| Control | 24.475 b | 0.375 a | 295.090 b | 187.340 a | 6.788 a | 30.483 c | 0.704 a |
| Drought | 20.123 b | 0.159 b | 198.400 c | 182.140 a | 4.578 b | 31.710 ab | 0.473 b |
| | | | 720 ppm | | | | |
| Control | 31.513 a | 0.312 a | 530.710 a | 184.980 a | 6.670 a | 31.263 b | 0.737 a |
| Drought | 24.475 b | 0.083 b | 291.480 b | 193.460 a | 2.498 c | 32.400 a | 0.406 b |
| Treatment [2,3] | ** | *** | *** | ns | *** | *** | *** |
| $CO_2$ | ns | * | *** | ns | * | ** | ns |
| Trt*$CO_2$ | ns | ns | *** | ns | * | ns | * |

[1] The measured intercellular $CO_2$/ambient $CO_2$ of LI-6400XT leaf cuvette. [2] Mean separation within the column by Duncan's multiple range test; ns, *, **, *** indicate non-significant or significant at $p \leq 0.05$, 0.01, 0.001, respectively; values followed by the same letter are not significantly different. Data are presented as means ± SE ($n = 10$). [3] SE-Standard error of the mean, Pn = 2.100; gs = 0.030; Ci = 13.200; ETR = 15.6; E = 0.400; Tleaf = 0.200; CiCa = 0.020.

The effect of DS on the fluorescence parameters is shown in Table 3. There was no interaction effect ($p > 0.001$) in all the fluorescence parameters. Furthermore, minimal fluorescence (Fo), maximal fluorescence (Fm), the quantum yield of photosystem II (ΦPSII), and photochemical quenching (qP) were not significantly affected under stress under both $CO_2$ levels, contradicting the previous report on basil and beech saplings [67,68]. The last statement also demonstrated that severe drought conditions decreased ATP and NADPH in photosynthetic metabolism and photorespiration and subsequently reduced the maximal quantum yield of photosystem II (FvFm) [69]. DS causes the suppression of photosynthesis activity which, in turn, reduces ΦPSII and increases the non-photochemical quenching (qN) [68]. On the contrary, qN FvFm, and ΦPSII decreased under the DS + $eCO_2$ in the present study when $eCO_2$ was applied. FvFm, qN, and ΦPSII showed a similar trend with Pn, Ci/Ca, Ci, and Tleaf. FvFm, ΦPSII, and qN are sensitive to DS and are not impacted significantly by the combination of $eCO_2$.

**Table 3.** The mean of light-adapted, minimal fluorescence (Fo'), dark-adapted, maximal fluorescence (Fm'), steady-state fluorescence (Fs), the maximal quantum yield of photosystem II photochemistry (Fv'/Fm'), the effective quantum yield of photosystem II photochemistry ($\Phi$PSII), the effective quantum yield of gas exchange measurements ($\Phi CO_2$), photochemical quenching (qP), and non-photochemical quenching (qN) of basil plants grown under without drought stress (control) and with drought stress at two levels of $CO_2$ (420 and 720 ppm) after 17 days of treatment.

| Treatment | Fo | Fm | Fs | Fv/Fm | $\Phi$PSII | $\Phi CO_2$ | qP | qN |
|---|---|---|---|---|---|---|---|---|
| | | | | 420 ppm | | | | |
| Control | 448.300 a | 840.500 a | 622.100 a | 0.466 b | 0.261 a | 0.0195 b | 0.558 ab | 1.875 b |
| Drought | 453.000 a | 815.400 a | 599.900 a | 0.444 b | 0.264 a | 0.0160 b | 0.593 a | 1.800 b |
| | | | | 720 ppm | | | | |
| Control | 440.800 a | 907.900 a | 674.100 a | 0.513 a | 0259 a | 0.0248 a | 0.507 b | 2.058 a |
| Drought | 457.600 a | 854.100 a | 638.900 a | 0.462 b | 0.249 a | 0.0163 b | 0.538 ab | 1.865 b |
| Treatment [1,2] | ns | ns | ns | * | ns | ** | ns | * |
| $CO_2$ | ns | ns | ns | * | ns | ns | ns | * |
| Trt*$CO_2$ | ns | ns | ns | ns | ns | ns | ns | ns |

[1] Mean separation within the column by Duncan's multiple range test; ns, *, ** indicate non-significant or significant at $p \leq 0.05$, 0.01, respectively; values followed by the same letter are not significantly different. Data are presented as means $\pm$ SE ($n = 10$). [2] SE-Standard error of the mean, Fo = 9.9; Fm = 34.5; Fs = 25.9; Fv/Fm = 0.01; $\Phi$PSII = 0.01; $\Phi CO_2$ = 0.001; qP = 0.03; qN = 0.05.

## 2.2. Carotenoid and Chlorophyll Analysis

Carotenoids contribute to photosynthesis and photoprotection in plants and are key metabolites for the proper functioning of photosynthetic apparatus during light intensity fluctuations [70–73]. Carotenoids ($\beta$-car) transfer the photochemical energy to chlorophyll to facilitate photosynthesis [74,75]. The basil plant subjected to DS shows a reduction in the $\beta$-car [41]. In our study, the individual major carotenoid pigments (Neoxanthin (Neo), Antheraxanthin (Anth), and Lutein (Lut)) were modulated under the DS, followed by $aCO_2$ and $eCO_2$ application (Figure 1). However, the application of $CO_2$ under DS did not modulate $\beta$-car (Figure 2). Neo showed the linear decreasing trend under DS + $eCO_2$. Neo concentrations of the control plant under $aCO_2$ were 276.4 ppm, which decreased to 206.9 ppm when applied under DS + $eCO_2$. On the other hand, there was no effect of DS and both $CO_2$ applications on Vio. However, Anth ($p < 0.001$) and Zea ($p < 0.001$) indicated a significant reduction in concentration under DS under both $CO_2$ applications compared to control. On the other hand, Lut showed a significant ($p < 0.001$) reduction in DS concentration with increased $CO_2$ concentration from $aCO_2$ to $eCO_2$. Thus, drought, on the one hand, reduced the concentrations of the pigments. On the other hand, the application of $CO_2$ accelerated the reduction of carotene pigments as the highest reduction was observed in Neo, Anth, Zea, and Lut under drought + $eCO_2$. It is reported that the effect of $eCO_2$ on carotenoids in leaves is different in different plants [73]. Some plants like *Solanum lycopersicum* and *Gyanura bicolor* increased while *Glycine max, Zea mays, Brassica napus*, and *Lactuca sativa* showed decreased carotenoids level under $eCO_2$ [73]. The present study also demonstrated that DS + $eCO_2$ application highly accelerated the decrease in all forms of carotene and xanthophylls, supported by the study by Dhami et al. [73]. Although Xanthophylls was unaffected by both $CO_2$ applications in control, its concentration reduced with increased $CO_2$ application (from $aCO_2$ to $eCO_2$) under DS, i.e., 370.2 to 314.1 ppm. Overall, Xanthophylls was reduced significantly ($p < 0.001$) under DS + $eCO_2$ (Table 2). The Za/Zav ratio decreased by 14.5–38.1% under the drought condition compared to control. Based on the result, we suggest that $eCO_2$ application caused the prominent decrease in carotenoids and xanthophylls, mainly under abiotic stress conditions [76–78].

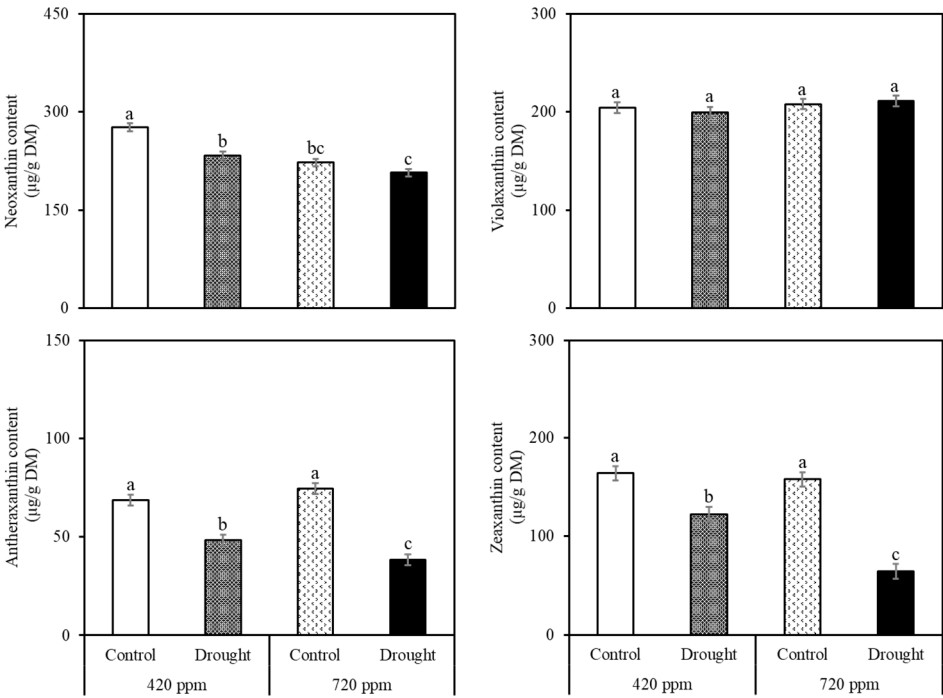

**Figure 1.** Neoxanthin (Neo), Violaxanthin (Vio), Antheraxanthin (Anth), and Zeaxanthin (Zea) estimation of basil plants grown without drought stress (control) and with drought stress at two levels of $CO_2$ (420 and 720 ppm) after 17 days of treatment. Data are presented as treatment means $\pm$ SE ($n = 10$). Different low case letters indicate a significant difference at $p < 0.05$ by the least significant difference. DM, Dry mass.

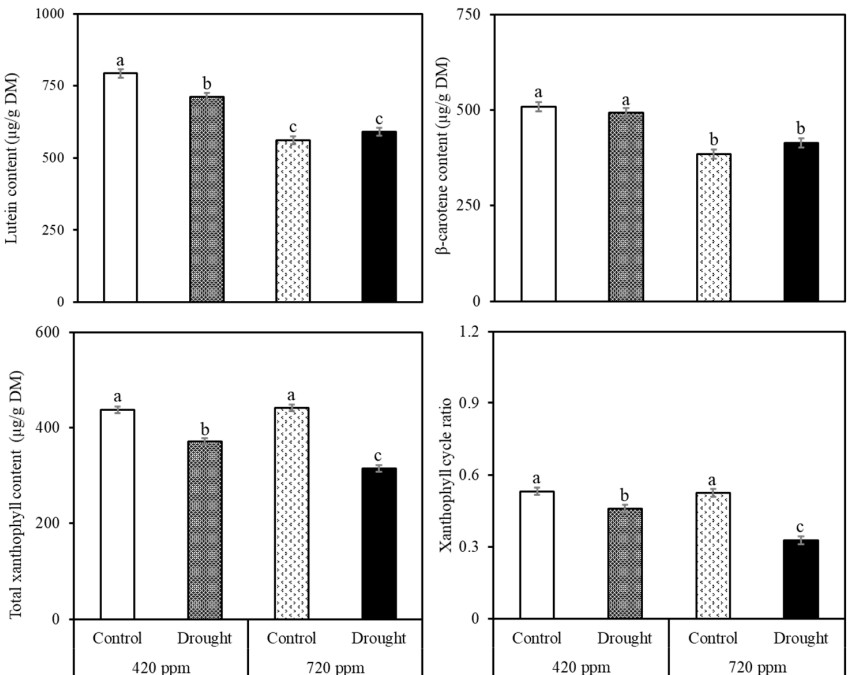

**Figure 2.** Lutein (Lut), β-carotene (β-car), Total xanthophyll, and Xanthophyll cycle ratio (Za/Zav) estimation of basil plants grown without drought stress (control) and with drought stress at two levels of $CO_2$ (420 and 720 ppm) after 17 days of treatment. Data are presented as treatment means $\pm$ SE ($n = 10$). Different low case letters indicate a significant difference at $p < 0.05$ by the least significant difference. DM, Dry mass.

Chlorophyll is an antioxidant and a signature pigment of photosynthetic organisms involved in photochemical activity [79,80]. Chlorophyll content in the plant is positively correlated to photosynthesis, and its reduction under abiotic stress like drought contributes to the inhibition of photosynthetic activity [41,81]. No interaction effect was observed in the present study on Chla, Chlb, and TChl (Figure 3). However, the treatment effect was observed on Chla and TChl, where Chla and TChl content significantly increased under DS, which contradicts the earlier reports [82,83], which could be because of $CO_2$ supplementation [84,85]. ChlB decreased in both drought and control conditions under $eCO_2$. The information further supports this result, with a significant decline of Chlb due to DS [86].

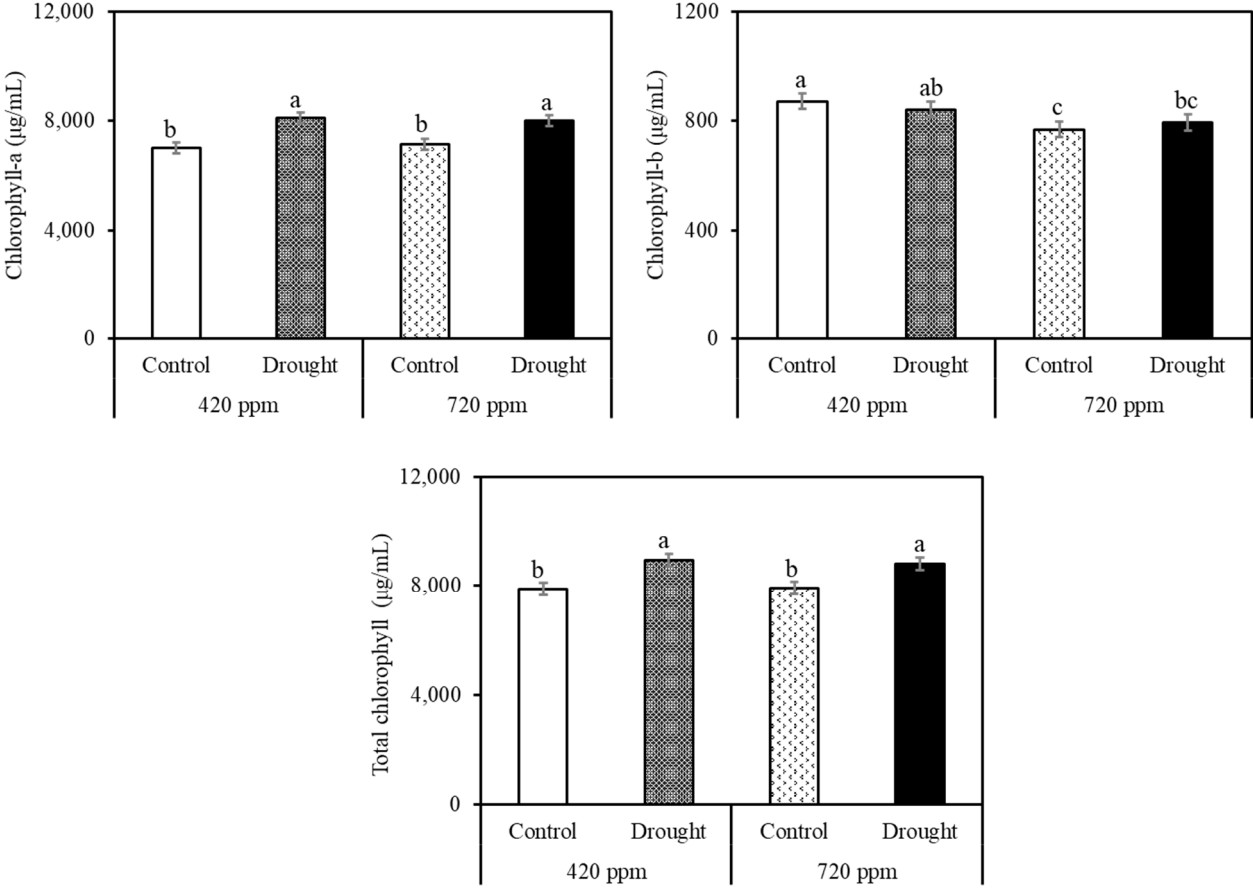

**Figure 3.** Chlorophyll-a (Chla), Chlorophyll-b (Chlb), and Total chlorophyll (TChl) estimation of basil plants grown without drought stress (control) and with drought stress at two levels of $CO_2$ (420 and 720 ppm) after 17 days of treatment. Data are presented as treatment means ± SE (*n* = 10). Different low case letters indicate a significant difference at *p* < 0.05 by the least significant difference.

### 2.3. Biochemical and Phytonutrient Analysis

The biochemical parameters measured are shown in Figure 4. There was no interaction effect in malondialdehyde (MDA), superoxide dismutase (SOD), trehalose, peroxidase, ascorbate, and glutathione under $CO_2$ levels. However, peroxidase and ascorbate increased under DS under both $CO_2$ groups compared to control. Peroxidase and ascorbate are crucial for acclimating a plant to any stress. Both peroxidase and ascorbate work together with SOD and catalase to protect the photosynthetic systems from oxidative damage by any environmental stress [87,88]. A previous report on $CO_2$ laser treatment revealed that $eCO_2$ enhances peroxidase and ascorbate, decreasing the MDA and $H_2O_2$, which strongly supports our study [89]. Furthermore, phenolic compounds are the metabolites responsible for antioxidant activity in basil, mainly produced in leaves and roots [90,91].

The present study demonstrated that there was a significant effect ($p < 0.001$) of $CO_2$ application on phenolics where it decreased under drought + $eCO_2$, which contradicted the result presented earlier in basil by Bekhradi et al. [92] and Al Jaouni et al. [44]. However, some metabolites tend to show species specificity, which might be the reason behind the decrease of phenolic compounds [44].

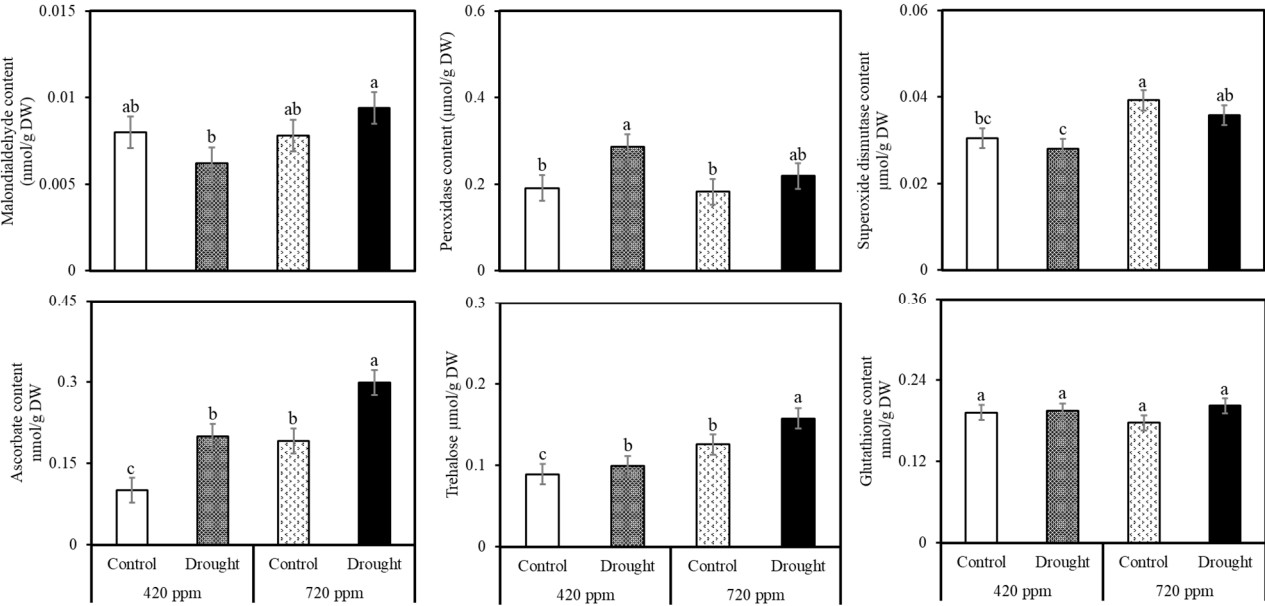

**Figure 4.** Malondialdehyde (MDA), peroxidase, superoxide dismutase (SOD), ascorbate, trehalose, and glutathione estimation of basil plants grown under without drought stress (control) and with drought stress at two levels of $CO_2$ (420 and 720 ppm) after 17 days of treatment. Data are presented as treatment means $\pm$ SE ($n = 10$). Different low case letters indicate a significant difference at $p < 0.05$ by the least significant difference.

## 3. Materials and Methods

### 3.1. Plant Materials and Growing Condition

Basil 'Genovese' seeds (Johnny's Selected Seeds, Winslow, ME, USA) were planted in polyvinyl chloride pots (15.2 cm diameter by 30.5 cm height) with a soil medium of 3:1 sand/soil classified as a sandy loam (87 percent sand, 2 percent clay, and 11 percent silt) and 500 g of gravel at the bottom of each pot. The experiment was set up under a randomized complete block design within a three-by-two factorial arrangement. Six SPAR chambers represent three blocks (10 replications each), where each chamber has 30 pots in three rows. More detailed information on the SPAR chamber was earlier elaborated by Reddy et al. [93] and Wijewardana et al. [94]. Basil plants were irrigated three times (700, 1200, and 1700 h) per day with full-strength Hoagland's nutrient solution via an automated computer-controlled drip system [95].

### 3.2. Treatments Application

Each chamber was maintained with 30/22 (day/night) °C temperature, with either ambient (420 ppm) or elevated (720 ppm) $CO_2$ concentrations. Temperature one hour after sunset was considered daytime temperatures, and one hour after sunset as nighttime temperatures. A full-strength Hoagland's solution [95] was applied to basil plants at 120 percent of evapotranspiration. For drought treatment, 50 percent of the full-strength Hoagland's solution was added to basil plants.

### 3.3. Physiological and Gas Exchange Measurements

The Dualex chlorophyll meter (FORCE-A, Orsay, France) was used to measure leaf chlorophyll (LChl), epidermal flavonoids (flav), epidermal anthocyanin (antho), and nitrogen balance index (NBI), with the device reader placed in the adaxial portion of the

leaf. Dualex is a hand-held leaf-clip tool that evaluates leaf quality using fluorescence and light transmission. For the OJIP fluorescence measurements on the second most completely developed leaf, a FluorPen FP 100 (Photon Systems Instruments, Drasov, Czech Republic) was utilized. At 17 DAT, the minimum fluorescence (Fo) was measured at 50 s when all PSII reaction centers were open, the maximum fluorescence (Fm) was measured when all PSII reaction centers were closed, and the steady-state fluorescence (Fs) was measured in each plant.

An LI-6400XT portable photosynthesis system (LiCor Biosciences, Inc., Lincoln, NE, USA) was used to measure photosynthesis and fluorescence parameters on the same leaf between 1000 and 1200 h at 18 DAT. The setup and regulation of relative humidity, the temperature of the chamber, and light intensity in the leaf chamber were followed as described by Barickman et al. [96]. Net photosynthesis (Pn) and quantum efficiency (Fv'/Fm') were measured when the total coefficient of variation (percent CV) was less than 0.5 percent. The device calculates transpiration rate (E), stomatal conductance (gs), internal $CO_2$ concentration (Ci), and electron transport rate (E) based on incoming and outgoing flow rates and leaf area (ETR). The interior to exterior $CO_2$ ratio was estimated using the Ci/Ca relationship.

### 3.4. Carotenoid and Chlorophyll Analysis

Carotenoid and chlorophyll pigments were extracted from freeze-dried basil tissues, according to Kopsell et al. [97], with a few modifications adopted by Barickman et al. [98].

## 4. Biochemical and Phytonutrient Parameters

### 4.1. Malondialdehyde (MDA)

Lipid peroxidation of membranes was estimated from MDA content, a final lipid peroxidation product, using the method described by Heath and Packer [99] with few modifications as described by Barickman et al. [96]. The extinction coefficient of 155 $mM^{-1}$ $cm^{-1}$ was used to determine the MDA concentration, which was reported as nmol $g^{-1}$ DW.

### 4.2. Hydrogen Peroxide ($H_2O_2$)

The content of $H_2O_2$ was measured following the method of Mukherjee and Choudhuri [100] with few modifications as described by Barickman et al. [96]. The content of $H_2O_2$ in samples was obtained from a standard curve using pure $H_2O_2$ and expressed as μmol $g^{-1}$ DW.

### 4.3. Superoxide Dismutase (SOD)

The activity of SOD was measured following the method of Dhindsa et al. [101] with few modifications by Awasthi et al. [102]. The reaction mixture (3 mL) used in this method has a mixture of 13 mM methionine, 25 mM nitro blue tetrazolium chloride (NBT), 0.1 mM EDTA, 50 mM sodium bicarbonate, 50 mM phosphate buffer (pH 7.8) and 0.1 mL enzyme extract. The absorbance of the samples was measured at 560 nm, and their total SOD activity was determined by evaluating their capacity to block the photochemical reduction of NBT. One unit of SOD activity was defined as the quantity of enzyme that inhibits the photochemical degradation of NBT by 50%. It was represented as SOD activity $mg^{-1}$ protein units.

### 4.4. Ascorbic Acid (ASC)

The estimation of ASC was done according to the combined method of Mukherjee and Choudhuri [100] and Awasthi et al. [102]. The ASC content was determined using a standard curve with a known ASC concentration and represented as nmoL $g^{-1}$ DW.

### 4.5. Trehalose

Trehalose concentration was estimated according to the method of Trevelyan and Harrison [103] and the Anthrone method of Brin [104]. Further details regarding the assay of an

enzyme associated with trehalose metabolism were described by Barickman et al. [96]. Trehalase activity was measured by activating phosphorylation with cAMP (cyclic adenosine monophosphate) and monitoring glucose levels [105].

### 4.6. Glutathione

Reduced glutathione was estimated according to the method of Griffith [106], adopted by Awasthi et al. [102]. Glutathione content was calculated from a standard graph calibrated by Griffith [106], and it was expressed as nmol $g^{-1}$ DW.

### 4.7. Data Analysis

SAS was used for statistical analysis on the data (version 9.4; SAS Institute, Cary, NC, USA), followed by PROC GLIMMIX analysis of variance (ANOVA) and mean separation. The study involved a randomized full block in a factorial arrangement with two water and two $CO_2$ treatments, three blocks, and ten replications. The standard errors were calculated using the ANOVA table's pooled error term. Duncan's multiple range test ($p < 0.05$) was used to distinguish between treatment classifications. Model-based values were reported rather than the unequal standard error from a data-based calculation because pooled errors reflect the statistical testing. To differentiate between treatment classifications, Duncan's multiple range test ($p < 0.05$) was used. Because pooled errors reflect statistical testing, model-based values were reported rather than the unequal standard error from a data-based calculation.

## 5. Conclusions

This study provided evidence that the interaction of DS and $eCO_2$ significantly impacts basil's overall physiological and biochemical aspects. $eCO_2$ positively impacted and increased the antho, and LChl by alleviating the adverse effect of DS. antho, on the other hand, decreased under DS + $aCO_2$ but increased under DS + $eCO_2$. LChl increased by 20% and 16% under DS + $aCO_2$ and DS + $eCO_2$, respectively. Furthermore, NBI increased under DS + $aCO_2$ by 26.2% compared to control. The application of $CO_2$ under DS did not modulate β-car. However, the individual primary carotenoid pigments (Neo, Anth, and Lut) were modulated under the DS, followed by $aCO_2$ and $eCO_2$ application. Neo showed the linear decreasing trend from under DS with the shift from $aCO_2$ (276.4 ppm) to $eCO_2$ (206.9 ppm). Anth and Lut decreased significantly under DS irrespective of both $CO_2$ treatments. Additionally, DS considerably inhibited the photosynthetic apparatus in plants by declining $CO_2$ availability and stomatal closure. Although $eCO_2$ could not increase Pn activity, gs and E under DS decreased with $eCO_2$ application, indicating that $eCO_2$ can uplift plants' water use efficiency by reducing the E and gs. FvFm, ΦPSII, and qN were sensitive to DS and were not impacted significantly by the $eCO_2$ application. Drought and $eCO_2$ accelerated most carotenes' reduction (Neo, Anth, Zea, and Lut) and Xanthophylls. It is worth addressing that $eCO_2$ supplementation can explain the increased chlorophyll content in basil under DS. Chlb decreased in both drought and control conditions under $eCO_2$. Peroxidase and ascorbate activity was higher due to the $eCO_2$ supply to acclimate the basil under the DS environment to withstand oxidative damage caused by $H_2O_2$. This study suggests that the application of $eCO_2$ during DS has a more significant impact on basil's photosynthetic parameters and biochemical traits than $aCO_2$.

**Author Contributions:** T.C.B.: conceptualization, methodology, validation, formal analysis, investigation, resources, data curation, writing—original draft, writing—review and editing, visualization, supervision, project administration, and funding acquisition. B.A.: formal analysis, writing—original draft, and writing—review and editing. A.S.: methodology, validation, and investigation. C.H.W.: methodology, validation, formal analysis, and investigation. K.R.R.: conceptualization, methodology, validation, formal analysis, investigation, resources, data curation, writing—review and editing, visualization, supervision, project administration, and funding acquisition. W.G.: conceptualization, methodology, validation, resources, and funding acquisition. All authors have read and agreed to the published version of the manuscript.

**Funding:** This material is based on the work supported by the USDA-NIFA Hatch Project under accession number 149210, and the National Institute of Food and Agriculture, 2019-34263-30552, and MIS 043050 funded this research.

**Institutional Review Board Statement:** Not applicable.

**Informed Consent Statement:** Not applicable.

**Data Availability Statement:** The data presented in this study are available on request from the corresponding author.

**Acknowledgments:** We thank David Brand for technical assistance and graduate students at the Environmental Plant Physiology Laboratory for their help during data collection. We would also like to thank Thomas Horgan for his technical help.

**Conflicts of Interest:** The authors declare that they have no known competing financial interests or personal relationships that could have appeared to influence the work reported in this paper.

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
