# Peer review of "Drought and Elevated CO2 Impacts Photosynthesis and Biochemicals of Basil (Ocimum basilicum L.)"

_stresses, doi:10.3390/stresses1040016_

Round 1

Reviewer 1 Report

This manuscript is about a Drought and Elevated CO2 Impacts Photosynthesis and Biochemicals of Basil (Ocimum basilicum L.). by T. Casey Barickman. It is an interesting article. I recommend some comments for this article.

Change the format of the article.

First introduction, second materials, and methods.

Separate results and discussion together because the article needs to separate and write more details of the discussion.

The conclusion was enough and good.

Results need more description about analysis.

My comments include:

After revised them, please send them back to me to review again 

My comments include:

Line 13 Increased atmospheric carbon dioxide (CO2), a major contributor to climate change, can compensate for drought-induced crop growth and production reductions

Line 40 Several other factors are causing DS,

Line 55  These changes directly and negatively impact soil temperature and moisture content, significantly decreasing crop yield during the next 50 years

Line 77 The respiration and ion uptake are reduced under DS, resulting in changes in crop metabolic and growth patterns and severe cases resulting in crop death

Line 94 Al Jaouni et al. [46] also reported that eCO (620 ppm) improved photosynthetic products' accumulation and biomass products by 40%. eCO also promotes the accumulation of phenolics, flavonoids, glutathione, and several other antioxidants to help combat damages from ROS during DS

Author Response

Please see attached response to reviewer comments.

Reviewer 2 Report

Review of Stresses1411812

Drought and Elevated CO2 Impacts Photosynthesis and Biochemicals of Basil (Ocimum basilicum L.)

  1. Casey Barickman , Bikash Adhikari , Akanksha Sehgal , C. Hunt Walne , K. Raja Reddy , Wei Gao

The authors wished to study the effects of drought and elevated CO2 on the physiology of basil and its synthesis of various pigments and secondary metabolites. They therefore grew basil under normal and semi-drought conditions, and under 420 and 720 ppm CO2 for a total of 4 treatments. They then measured various photosynthetic parameters of plants grown under these 4 conditions, including net photosynthesis, conductance, transpiration rate and various fluorescence parameters. As might be expected, drought stress severely reduced conductance, transpiration and Ci, and somewhat reduced net photosynthesis. Elevated CO2 helped reduce the impact of drought stress on the photosynthetic parameters, and on the production of various pigments and secondary metabolites.

This general topic has been studied several times previously in basil in addition to many other plants, although under different conditions. In particular, the pCO2 and drought treatments differed, which might partly account for the differences with prior results. It’s also unclear whether the same cultivars were used as in previous studies.

These points aside, this report does provide a great deal of useful data. It seems to have been performed competently using suitable techniques and with adequate replication. I therefore recommend that it be considered for publication after correcting the problems listed below.

Concerns

How well did the plants grow under these different conditions? How did these treatments affect the overall productionof biomass?

There are too many unexplained acronymns such as eCO2 , E, Pn, etc in the abstract and main text. Always spell them out the first time that they are mentioned.

All tables should list means ± SE, rather than giving SE in caption, and far too many significant figures are given. If the SE is ± 0.9, there is one significant figure after the decimal point, not 3! State in the caption that the data are means ± SE, the number of replicates, and the test used to determine statistical significance.

Do not list SEs in figure captions. Just state that bars are means ± SE, and state the number of replicates.

Abstract should state what the experimental conditions and CO2 levels were.

It’s a bit of an overstatement to claim that basil makes a major contribution to global food security. It has an important role, but nowhere near the level of rice, maize or wheat, not to mention beans and potatoes amongst many others!

Specific comments

Lines 18-20 are hard to understand and should be rewritten for clarity

Lines 23-25 are hard to understand and should be rewritten for clarity

Lines 31-33 please correct mistakes

Lines 62-63 are hard to understand and should be rewritten for clarity. Are you saying that Basil is the most wide-spread warm season aromatic and medicinal herb within this family, or that it is the most wide-spread warm season aromatic and medicinal herb, and it belongs to this family?

Line 70: You already defined drought stress on line 39

Lines 77-79 are hard to understand and should be rewritten for clarity

Lines 91-94 should be rewritten for clarity

Line 111: explain that these chemicals were measured in basil grown in all four conditions.

Lines 116-117 are hard to understand and should be rewritten for clarity

Line 136: should Table 7 be table 2?

Lines 172-173 are hard to understand and should be rewritten for clarity

Lines 188-195 are hard to understand and should be rewritten for clarity

Lines 210-213 are hard to understand and should be rewritten for clarity

Lines 247-250 are hard to understand and should be rewritten for clarity

Lines 276-277 should be rewritten for clarity

Line 280: does this mean that drought-stressed plants were watered at 60% of evapotranspiration?

Lines 289-291 are hard to understand and should be rewritten for clarity

The entire conclusions section should be rewritten for clarity

Author Response

Please see attached response to reviewers comments.

Round 2

Reviewer 1 Report

Dear Authors

I accept all your revision. Well done.

Best regards

Reviewer 2 Report

Review of revised Stresses1411812

Drought and Elevated CO2 Impacts Photosynthesis and Biochemicals of Basil (Ocimum basilicum L.)

  1. Casey Barickman , Bikash Adhikari , Akanksha Sehgal , C. Hunt Walne , K. Raja Reddy , Wei Gao

The authors have addressed most of my concerns.

However, they still give the SE of their data in the captions, rather than in the tables. This is not acceptable, and must be corrected! List the data in the tables as means ± SE! Also, please use appropriate sig figs. How can a value be 4.578 when the SE is ± 0.4? This must be corrected!

Moreover, they should refer to their previous publications to state how these treatments affected growth and productivity, and in their discussion they should address which of the changes in physiology that they measured might have caused the reductions in growth and productivity. This should then be discussed in terms of the most useful ways to increase basil productivity in light of the projected changes in eCO2, temperature and water availability.

The conclusions section is improved, but is still hard to understand. Please state in simple sentences what you learned frm this study and how this information can be used to improve basil farming and farming in general.

In addition, there are still numerous minor problems with the English that should be corrected before publication.

A general comment, please pay attention to singular versus plural. For example, on line 119 “Both eAntho and eFlav is reported” should be ”Both eAntho and eFlav are reported”. This paper is full of similar errors.

Also, please use the appropriate tense for past, present or future. Since you are reporting work that you did, most of the paper should be written in the past tense. For example line 194 states that “The basil plant subjected to DS shows a reduction” This paper is full of similar errors.

Line 16 should explain the purpose of the experiment before describing how it was set up.

Lines 17-18 are hard to understand and should be rewritten for clarity

Lines 40-42 are hard to understand and should be rewritten for clarity. Please explain that it is lack of water that often limits plant productivity.

Line 71: what does “exceptionally used” mean? Please rewrite!

Lines 82-84 are hard to understand and should be rewritten for clarity.

Lines 96-97 are hard to understand and should be rewritten for clarity.

Line 114: please explain why you measured these compounds?

Lines 146-149 are hard to understand and should be rewritten for clarity and grammar.

Lines 157-159 are hard to understand and should be rewritten for clarity.

Lines 198-203 are hard to understand and should be rewritten for clarity.

Lines 215-218 are hard to understand and should be rewritten for clarity.